# Pairing patterns in one-dimensional
# spin- and mass-imbalanced Fermi gases

**Lukas Rammelmüller[1,2,3]⋆, Joaquín E. Drut[4] and Jens Braun[2,5,6]**

**1** Arnold Sommerfeld Center for Theoretical Physics, University of Munich,
Theresienstr. 37, 80333 München, Germany
**2** Institut für Kernphysik (Theoriezentrum), Technische Universität Darmstadt,
Schlossgartenstraße 2, D-64289 Darmstadt, Germany
**3** GSI Helmholtzzentrum für Schwerionenforschung GmbH,
Planckstraße 1, D-64291 Darmstadt, Germany
**4** Department of Physics and Astronomy, University of North Carolina,
Chapel Hill, North Carolina 27599, USA
**5** FAIR, Facility for Antiproton and Ion Research in Europe GmbH,
Planckstraße 1, D-64291 Darmstadt, Germany
**6** ExtreMe Matter Institute EMMI, GSI,
Planckstraße 1, D-64291 Darmstadt, Germany

⋆ lukas.rammelmueller@physik.uni-muenchen.de

## Abstract

We study spin- and mass-imbalanced mixtures of spin-$\frac{1}{2}$ fermions interacting via an attractive contact potential in one spatial dimension. Specifically, we address the influence of unequal particle masses on the pair formation by means of the complex Langevin method. By computing the pair-correlation function and the associated pair-momentum distribution we find that inhomogeneous pairing is present for all studied spin polarizations and mass imbalances. To further characterize the pairing behavior, we analyze the density-density correlations in momentum space, the so-called shot noise, which is experimentally accessible through time-of-flight imaging. At finite spin polarization, the latter is known to show distinct maxima at momentum configurations associated with the Fulde-Ferrell-Larkin-Ovchinnikov (FFLO) instability. Besides those maxima, we find that additional features emerge in the noise correlations when mass imbalance is increased, revealing the stability of FFLO-type correlations against mass imbalance and furnishing an experimentally accessible signature to probe this type of pairing.

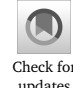

# 1 Introduction

Ultracold quantum gases represent excellent test grounds to challenge our understanding of the physics of strongly interacting Fermi systems. The unprecedented experimental plasticity of these systems allows for the investigation of a rich variety of physical scenarios in a clean and controllable way [1–4]. An intriguing phenomenon occurs in attractively interacting spin-$\frac{1}{2}$ Fermi gases, where singlet pair-formation of spin-up and spin-down particles leads to superfluid behavior at low temperatures. In the case of equally populated spin species, the mechanism is well understood and can be explained via conventional Bardeen-Cooper-Schrieffer (BCS) theory.

Our understanding of fermion pairing can be further tested by considering deviations from the balanced limit of equal Fermi surfaces. A prominent example for such a "deformation" is obtained by introducing a spin imbalance. In this case, superfluidity is generally expected to break down above a critical polarization, commonly referred to as the Chandrasekhar-Clogston limit [5,6]. Below the critical polarization, so-called polarized superfluid states may be stabilized, for which several different mechanisms have been proposed in the literature [7–9]. A particularly intriguing scenario was put forward independently by Fulde and Ferrell (FF) and by Larkin and Ovchinnikov (LO), who argued on very general grounds that a spatially inhomogeneous ground state may be energetically most favorable even in infinite systems [10,11]. In this case, the emerging pairs of spin-up and spin-down fermions carry a finite total momentum, potentially giving rise to the formation of a spatially varying order parameter, generally referred to as the FFLO state (see, e.g., Refs. [12,13] for an illustration).

Despite the experimental ability to study population-imbalanced Fermi gases [14–17], a smoking-gun evidence for the spontaneous formation of FFLO-type pairing is still missing. In addition to the increased technical challenge, part of the reason for the absence of a clear signal of inhomogeneous pairing is the "lack of stability" of this inhomogeneous phase: For three-dimensional (3D) systems, beyond mean-field calculations suggest that only a thin layer of the parameter space between the homogeneous superfluid and the normal fluid is occupied by an inhomogeneous phase, if at all [13, 18–21]. For one dimensional (1D) systems, on the other hand, inhomogeneous pairing is expected to exist in a wide range of parameter space [22–25]

and experimental measurements are indeed consistent with FFLO-type pairing [16,17].

On the theory side, continuum 1D systems with zero-range interaction are described by the so-called Gaudin-Yang model which can be solved in closed form via the famous Bethe ansatz (BA) [4, 26, 27]. While many ground-state properties are accessible within this approach, correlation functions remain notoriously challenging to compute and typically one has to either resort to low-energy effective treatments, such as the Tomonaga-Luttinger-liquid (TLL) description, or use numerical methods, such as the density-matrix renormalization group (DMRG), exact diagonalization (ED) or Monte Carlo (MC) approaches.

In addition to the preparation of systems with two different hyperfine states of the same atom species, recent experiments explored the possibility to realize heteronuclear mixtures such as $^6$Li-$^{40}$K, $^{40}$K-$^{161}$Dy or $^6$Li-$^{53}$Cr, which feature a mass imbalance between the spin-up and spin-down fermions [28–31]. These setups represent a compelling test ground for our picture of the underlying pair formation. However, their theoretical description remains relatively scarce in the literature since the reduced symmetry of the model renders BA studies inapplicable, except for a very few specific values for the mass ratio. Nevertheless, progress on analytic insights has been made for special symmetric configurations in the few-body sector [32–34] and numerically the regime was explored by worldline MC [35,36], ED [37] and complex Langevin (CL) studies [38,39]. Beyond the few-body regime, several studies based on effective descriptions have been conducted [40–44] and numerical studies investigated the phase-diagram of the asymmetric 1D Hubbard model (which coincides with the continuum model at low fillings) by means of DMRG [45,46] as well as spin- and mass-imbalanced systems via MC [47], time-evolving block decimation (TEBD) method [48] and DMRG [49] approaches.

The present study aims to complement the above investigations by exploring two-body correlation functions for general spin and mass imbalances for attractive interactions. Our ultimate goal is to arrive at an experimentally accessible quantity to study pairing in spin- and mass-imbalanced systems, and to that end we present numerical results for the so-called shot-noise correlation function as obtained from the CL approach [50–52]. The latter was recently found to provide a promising way to circumvent the sign problem in nonrelativistic Fermi gases [38,53,54]. It is an additional objective of the present work to further develop the method in this context by presenting the first study of spin- and mass-imbalanced systems in the strict $T = 0$ limit via a projective formulation, as well as the first CL determination of two-body correlation functions for nonrelativistic systems. The CL method is a versatile tool which, unlike other approaches mentioned above, may be readily extended to higher spatial dimensions. Indeed, this capability was recently demonstrated for 3D fermions at unitarity, where excellent agreement with state-of-the-art numerical and experimental data for the equation of state was reported [55].

The work is organized as follows: In Sec. 2, we introduce the model, define the relevant scales, and briefly discuss some aspects of the CL approach; a detailed discussion of our CL approach to ultracold Fermi gases can be found in Refs. [38, 54, 56, 57]. Numerical results are subsequently presented, where we first validate our numerical study via a comparison of ground-state energies to known exact results obtained with the BA method. We then proceed with the main focus of the present work, namely the computation of two-body correlation functions of the purely spin-polarized case in Sec. 3, and finally of general spin- and mass-imbalanced systems in Sec. 4. Our conclusions and a brief outlook are presented in Sec. 5.

## 2 Model, Scales & Method

### 2.1 Model and Scales

We consider systems of two fermionic species interacting via contact interaction in a one-dimensional periodic box of extent $L$. Ignoring interactions between fermions of the same species, the associated Hamilton operator in its second-quantized form reads

$$\hat{H} = \int dx \left\{ \sum_{\sigma=\uparrow,\downarrow} \hat{\psi}_\sigma^\dagger(x) \left( -\frac{\hbar^2 \nabla^2}{2m_\sigma} \right) \hat{\psi}_\sigma(x) + g \hat{\psi}_\uparrow^\dagger(x) \hat{\psi}_\uparrow(x) \hat{\psi}_\downarrow^\dagger(x) \hat{\psi}_\downarrow(x) \right\}, \qquad (1)$$

where $\hat{\psi}_\sigma^\dagger$ ($\hat{\psi}_\sigma$) represent creation (annihilation) operators for a fermion of species $\sigma$ and mass $m_\sigma$. In the following, units are chosen such that $\hbar = k_B = 1$. In the mass-balanced case, we set $m_\uparrow = m_\downarrow = 1$. For our conventions in the case of unequal masses, we refer to Sec. 4. Note that at low densities this system corresponds to the (asymmetric) 1D Hubbard model.

In our numerical studies below, we always work in the canonical ensemble which implies that the particle numbers $N_\sigma$ of the spin-up and spin-down fermions are parameters at our disposal, determining the densities $n_\sigma = N_\sigma / L$ of the two species and their Fermi momenta $k_F^\sigma = \pi n_\sigma$. Assuming that discretization artifacts in our CL studies on a finite lattice are negligible, physical observables depend on three parameters, namely the mass ratio $\kappa = m_\uparrow / m_\downarrow$, the spin polarization $p$,

$$p \equiv \frac{N_\uparrow - N_\downarrow}{N_\uparrow + N_\downarrow}, \qquad (2)$$

and the dimensionless coupling $\gamma = g/n$ with $n = n_\uparrow + n_\downarrow$ being the total density. The latter can be related to the s-wave scattering length $a_s$ via $g = 2/a_s$ (see, e.g., Ref. [58]).

A meaningful comparison of systems associated with different mass ratios $\kappa$ requires to identify one scale which is kept fixed to the same value in all systems. In this work, we shall keep the two-body binding energy $\varepsilon_B$ fixed for $g < 0$ (attractive case):

$$\varepsilon_B = -\frac{\kappa}{2(1+\kappa)} g^2. \qquad (3)$$

In practice, this is conveniently achieved by introducing a suitably chosen rescaled coupling constant $\tilde{\gamma}$:

$$\tilde{\gamma} = \sqrt{\frac{2\kappa}{1+\kappa}} \gamma. \qquad (4)$$

Thus, for fixed $\tilde{\gamma}$, the dimensionless binding energy $\varepsilon_B/n^2 = -\tilde{\gamma}^2/4$ then remains constant when we vary the mass ratio $\kappa$. Note that we have $\tilde{\gamma} = \gamma$ in the mass-balanced limit $\kappa = 1$.

### 2.2 Outline of the numerical treatment

For our studies of ground-state properties of spin- and mass-imbalanced Fermi gases, we start from the canonical partition function $\mathcal{Z}$ which is obtained by projecting an arbitrary trial state $|\psi_T\rangle$ (typically chosen to be a Slater determinant) onto the ground state in the limit of large imaginary times $\beta$:

$$\mathcal{Z}_\beta = \langle \psi_T | e^{-\beta \hat{H}} | \psi_T \rangle. \qquad (5)$$

This partition function (not to be confused with its finite temperature counterpart, whose calculation requires summing over a complete set of states) can be rewritten in terms of a Euclidean-time path integral over a Hubbard-Stratonovich (HS) field $\phi$, which is introduced

to integrate out the fermions exactly, yielding the usual expression in terms of fermionic determinants

$$\mathcal{Z}_\beta = \int \mathcal{D}\phi \; \det M_\uparrow[\phi, \beta] \det M_\downarrow[\phi, \beta]. \tag{6}$$

This expression is the basis of the well-known class of auxiliary-field methods [59]. The path integral in $\mathcal{Z}_\beta$ can now be tackled straightforwardly with conventional stochastic methods, provided that the system is in a mass- and spin-balanced state. In that case, the determinants of the two fermion species are equal and their product yields a nonnegative number. Away from this balanced configuration, a standard MC evaluation of the path integral (6) is complicated by the infamous sign problem [54]. In order to tackle the systems at the heart of this work, we therefore employ the CL approach which allows us to surmount the sign problem by a complexification of the HS field [54, 60]. The key ingredient is the use of a complexified version of the Langevin equation to produce a Markov chain of randomly distributed field values $\phi$ which then allows us to measure observables in the same fashion as in conventional MC approaches based on, e.g., the Metropolis algorithm. Nevertheless, it should be stressed that the CL approach is still a method under construction and this prescription is not without shortcomings in certain theories, as discussed in Refs. [61–64]. As remarked above, however, these potential issues seem to be under control for ultracold Fermi gases in a wide parameter range [54], in particular for attractive interactions [36, 38].

In practice, the field integration of Eq. (6) is performed by discretizing spacetime (on which the field $\phi$ lives) into a spacetime lattice of size $N_x \times N_\tau$, where $N_x$ and $N_\tau$ determine the number of grid points in spatial and imaginary time direction, respectively. Thus, the field integral in Eq. (6) becomes an integral over a finite (albeit large) number of dimensions. In our computations of the pair-correlation functions, and their corresponding pair-momentum distribution functions and ground-state energies, we have fixed the density and polarizations in lattice sizes in the range $N_x = 120 - 140$. Such lattice sizes allow for good resolution of the correlation functions and result in negligible finite-size effects. A discussion of finite-size effects of similar quantities can be found in Ref. [65]. Moreover, we fixed the spatial lattice constant to $\ell = 1$, such that the first Brillouin zone extends from $-\pi$ to $\pi$. For the imaginary-time axis it was found sufficient to use a discretization step of $\Delta\tau = 0.05$. In order to reach the ground state, statistically independent runs were performed for different projection times corresponding to $N_\tau = 50 - 150$ (depending on the density). Averaging runs across multiple converged projection times then yields robust estimates of ground-state properties, see, e.g., Ref. [66].

Our sampling strategy consists in integrating the CL equation for the HS field with a suitably chosen step size up to trajectory lengths $t_L \approx 1000$. Here, $t_L$ denotes the fictitious Langevin time that parametrizes the Markov chain and relates to the number of samples $N$ via $t_L = N \Delta t_L$. The parameter $\Delta t_L$ denotes the adaptive integration step of the discrete CL equations and has been fixed to an average value of $\Delta t_L^{(0)} = 0.04$ which was found to be sufficient for the study of the correlation functions presented below. Moreover, we have used a regulator term of strength $\xi = 0.1$ in order to stabilize the CL trajectories, as also done in previous studies with this method [38, 39, 56]. We have checked that a further reduction of this parameter leaves our present results for the correlation functions unaltered, such that an extrapolation to the limit $\xi \to 0$ is not needed.

During the evolution of the random process we take "snapshots" of the HS field separated by $\sim 1.0$ "Langevin seconds" which is sufficient to yield decorrelated samples. Unless specified otherwise, data points reflect averages over five such CL trajectories associated with randomly chosen initial conditions of the CL equation. This yields roughly 5000 decorrelated samples in total, allowing for a statistical uncertainty of $\sim 1 - 2\%$. At this point, we would like to emphasize that the error bars presented in this work reflect purely statistical uncertainties and

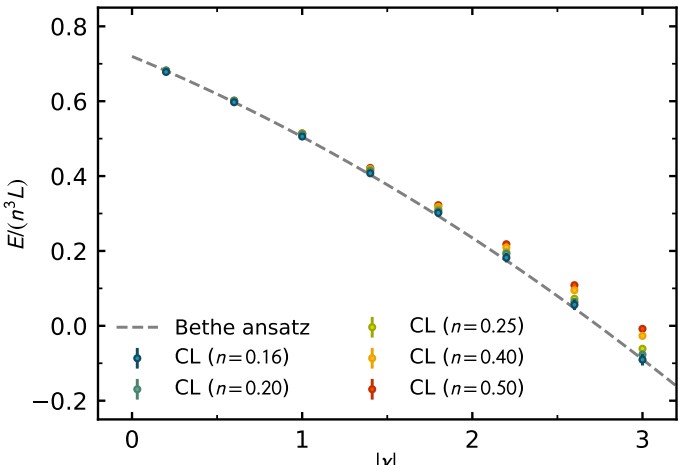

Figure 1:  Ground-state energy for attractively interacting spin-up and spin-down fermions in units of $n^3 L$ as a function of the modulus of the coupling strength $\gamma$ as obtained from our CL study (symbols). Systems with fixed spin-polarization $p = 0.5$ for total densities of $n = 0.16, 0.2, 0.25, 0.4$ and $0.5$ are shown. For comparison, the exact BA result for the thermodynamic limit [67] is shown (gray dashed line).

do not contain systematic artifacts associated with the discretizations underlying our numerical solution of the CL equation.

Finally, we note that we have monitored running averages and histograms of computed observables to analyze the reliability of our calculations. For all systems discussed in the present work, we have found well-behaved distributions with well-defined second moments and most importantly without so-called fat tails that could spoil the expectation values, see also Ref. [65] for a related discussion. Further checks, specific to the CL approach, include the investigation of imaginary parts of computed quantities which we have found to be exponentially decaying with increasing $t_{\mathrm{L}}$ as expected for physical observables.

## 3   Pairing in population-imbalanced Fermi gases

The main focus of this section is the investigation of pairing with the aid of appropriate two-body correlation functions. To this end, we consider systems with a large number of particles on large lattices, i.e., systems sufficiently close to the thermodynamic limit (TL) where finite-size effects should be negligible. Moreover, to be consistent with our previous work [65], we only consider odd numbers of particles for a given species to ensure that we do not explicitly break translational invariance.

### 3.1   Ground-state energy

As a validation of our numerical framework, it is instructive to first look at ground-state energies and compare them with exact results from the literature. In Fig. 1, we show the ground-state energy for systems with fixed polarization $p = 0.5$ at various densities as a function of the modulus of the coupling strength $\gamma$ with $\gamma < 0$ (attractive case). A comparison with the exact solution [67] reveals significant deviations from the thermodynamic limit (TL) for systems with $n = 0.5$ and $|\gamma| = 3.0$. This is a consequence of the discretization of the spatial volume in our study. In fact, the large density induces a finite interaction range on the lattice which causes a deviation from the pure zero-range limit in the continuum. However, we find

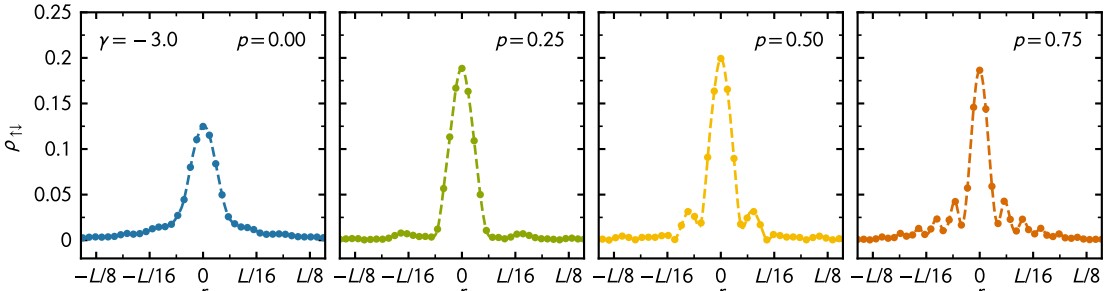

Figure 2: On-site pair-correlation function for systems of fixed density $n = 0.4$ and polarizations of $p = 0.0, 0.25, 0.5$ and $0.75$ (from left to right). The "oscillatory behavior" with increasing polarization is clearly visible. Note that the envelope of the correlation functions obeys a power-law behavior for large $r$. The statistical error bars are smaller than the symbol size.

convergence to exact results with decreasing density. For dilute systems with $n = 0.16$, we find excellent agreement across the considered range of interaction strengths, indicating the validity of our numerical approach.

## 3.2 On-site pair-correlation function

A prominent difference between 1D and 3D systems is the effect of long-range fluctuations which prohibit the spontaneous breakdown of a continuous symmetry in the 1D case [68,69]. Instead, quasi long-range order (QLRO) takes place [70–72], manifesting itself in a algebraic decay of two-body correlation functions:

$$C(r) \propto r^{-\Delta_C} . \tag{7}$$

Here, $r = |x - x'|$ and $\Delta_C$ denotes the correlation exponent associated with the underlying ordering mechanism. The dominant instability is the one with the smallest correlation exponent, i.e., the one whose correlations survive the longest distances. For spin-imbalanced Fermi gases away from half-filling, it has been found that spin-singlet pairing is dominant in the parameter range of interest in this work [23]. For this reason, we compute the on-site pair-correlation function, defined as the overlap of a state with a point-like pair of spin-up and spin-down fermions removed at point $x$ from the ground state and a state with such a pair removed at point $x'$ from the ground state:

$$\rho_{\uparrow\downarrow}(x, x') \equiv \langle \hat{\psi}_\uparrow^\dagger(x') \hat{\psi}_\downarrow^\dagger(x') \hat{\psi}_\downarrow(x) \hat{\psi}_\uparrow(x) \rangle. \tag{8}$$

In addition to the algebraic decay of this correlation function, spin-imbalanced systems are expected to exhibit an additional feature at sufficiently large distances $r = |x - x'|$, namely a spatially oscillating behavior of the form [72]:

$$\rho_{\uparrow\downarrow}(x, x') \propto |\cos(q|x - x'|)| \, |x - x'|^{-\Delta_{\uparrow\downarrow}} . \tag{9}$$

Here, $q = |k_F^\uparrow - k_F^\downarrow|$ quantifies the population difference and $\Delta_{\uparrow\downarrow}$ is the correlation exponent associated with the pair-correlation function. Since the long-range behavior of $\rho_{\uparrow\downarrow}$ is related to the order parameter for the fermion gap, a phenomenological interpretation of the minima of $\rho_{\uparrow\downarrow}$ can be given [72]: These minima may be viewed as "pockets" in which the excess fermions (here: excess spin-up fermions for $p > 0$) tend to reside since the positions of these minima are associated with vanishing (or small) fermion gap.

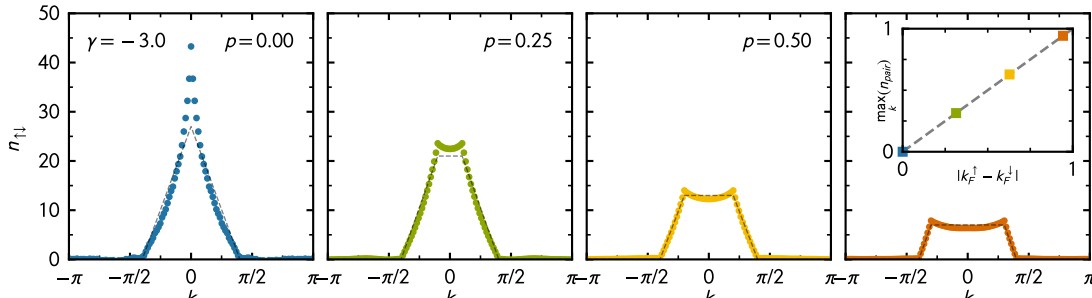

Figure 3: Pair-momentum distribution $n_{\uparrow\downarrow}$ in the first Brillouin zone for systems of fixed density $n = 0.4$ and polarizations of $p = 0.0, 0.25, 0.5$ and $0.75$ (from left to right). A clearly visible maximum/peak away from the point $k = 0$ emerges for any non-zero polarization. The gray dashed lines reflect the non-interacting system. **Inset of rightmost panel**: The dependence of the position of the maximum (for $k > 0$) on the polarization in form of the difference $|k_F^{\uparrow} - k_F^{\downarrow}|$ is found to be in excellent agreement with the FFLO prediction. The statistical error bars are smaller than the symbol size.

In Fig. 2, we show our results for the on-site pair-correlation function for a total density $n = 0.4$ and for different values of the polarization $p$. In the unpolarized limit, the monotonic decay of the pair-correlation function is apparent, whereas oscillatory behavior is observed for any finite polarization. Moreover, the frequency of the oscillations increases when the polarization increases, as suggested by Eq. (9). We observe the above behavior for all polarizations considered in this work, indicating the absence of a Chandrasekhar-Clogston limit in 1D, in agreement with earlier MC and DMRG studies of the spin-imbalanced 1D Hubbard model [23,73].

To further analyze pairing, it is instructive to study the pair-momentum distribution which is the Fourier transform of Eq. (8):

$$
\begin{aligned}
n_{\uparrow\downarrow}(k, k') &= \sum_{x,x'} \varphi_k^*(x) \rho_{\uparrow\downarrow}(x, x') \varphi_{k'}(x') \\
&= \sum_{p,p'} \left\langle \hat{\chi}_{\uparrow}^{\dagger}(k) \hat{\chi}_{\downarrow}^{\dagger}(k-p) \hat{\chi}_{\downarrow}(k') \hat{\chi}_{\uparrow}(k'-p') \right\rangle .
\end{aligned}
\tag{10}
$$

Here, we have introduced the creation and annihilation operators in momentum space:

$$
\hat{\chi}_{\sigma}(k) = \sum_x \varphi_k(x) \hat{\psi}_{\sigma}(x), \quad \hat{\chi}_{\sigma}^{\dagger}(k) = \sum_x \varphi_k^*(x) \hat{\psi}_{\sigma}^{\dagger}(x),
\tag{11}
$$

with the the box-normalized single-particle orbitals

$$
\varphi_k(x) = \frac{1}{\sqrt{L}} e^{i \frac{2\pi k}{L} x}.
\tag{12}
$$

The pair-momentum distribution quantifies the probability of finding a zero-size (i.e. point-like) pair composed of one spin-up and one spin-down fermion with a given momentum in the system. Thus, the FFLO state reveals itself through off-center peaks at $k = \pm q$. Of course, from our analysis of the on-site pair-correlation function, the appearance of these off-center peaks does not come unexpected. Indeed, if pairing predominantly occurs at non-zero momenta, the corresponding real-space signal will be modulated by the appropriate wavenumber.

In Fig. 3 we show results for the diagonal part of the pair-momentum distribution, i.e., $n_{\uparrow\downarrow}(k, k)$, at fixed density $n = 0.4$ for various spin imbalances. For the spin-balanced case, we

observe a pronounced peak in $n_{\uparrow\downarrow}$ at $k = 0$, as it is expected from BCS theory. This is also in agreement with previous MC studies of such systems [65]. The situation changes as soon as we turn on a finite spin imbalance. We then observe peaks at $k \neq 0$ as a consequence of pairing across mismatched Fermi points. To illustrate this, we show the pair-momentum distribution for systems with polarizations $p = 0.25, 0.5, 0.75$ in Fig. 3. The position of the off-center peaks in momentum space follow the prediction for FFLO type pairing. This is highlighted in the inset of the right panel of Fig. 3, where the dependence of the peak positions as a function of the mismatch $q = |k_F^{\uparrow} - k_F^{\downarrow}|$ is depicted.

## 3.3 Noise correlations

In addition to the pair correlator, we also discuss the density-density correlation function in momentum space – often referred to as the "atomic shot-noise". It is defined as follows

$$
\begin{aligned}
G_{\sigma\sigma'}(k, k') &\equiv \langle \Delta \hat{n}_{k,\sigma} \, \Delta \hat{n}_{k',\sigma'} \rangle \\
&= \langle \hat{n}_{k,\sigma} \, \hat{n}_{k',\sigma'} \rangle - \langle \hat{n}_{k,\sigma} \rangle \langle \hat{n}_{k',\sigma'} \rangle.
\end{aligned}
\tag{13}
$$

Here, $\hat{n}_{k,\sigma}$ denotes the number operator in momentum space for the species $\sigma$ and $\Delta \hat{n}_{k,\sigma} = \hat{n}_{k,\sigma} - \langle \hat{n}_{k,\sigma} \rangle$ denotes the fluctuation around the thermal average.

This quantity has been proposed as a suitable probe for correlations in ultracold quantum gases [74]. Several theoretical studies predict distinct imprints of the underlying ordering mechanism in the shot noise [23, 75–77]. Experimentally, it is accessible through the analysis of time-of-flight images of the spatial density profile after ballistic expansion, i.e., after interactions and the trap have been switched off. The so-obtained time-of-flight images are then proportional to the single-particle momentum distribution. A requirement for the experimental measurement is the ability of spin-selective imaging of the sample density which has been applied in several setups so far (see, e.g., Refs. [78–80]).

Loosely speaking, the computation of $G_{\uparrow\downarrow}(k, k')$ allows us to study the internal momentum structure of a pair of one spin-up and one spin-down fermion. Indeed, we can combine the momenta $k$ and $k'$ such that we obtain $k_{\text{tot}} = (k + k')$ and $k_{\text{rel}} = k - k'$ being the total and relative momentum of the two fermions in a pair, respectively. To understand the quantity itself, it is instructive to think of it as the covariance matrix of the single-particle momentum distributions $n_{\sigma}(k)$ and $n_{\sigma'}(k')$: Positive values of $G_{\sigma\sigma'}(k, k')$ correspond to situations where high (low) values of $n_{\sigma}(k)$ occur alongside with high (low) values of $n_{\sigma'}(k')$, whereas negative $G_{\sigma\sigma'}(k, k')$ corresponds to anti-correlation, i.e. high (low) values of $n_{\sigma}(k)$ entail low (high) values of $n_{\sigma'}(k')$. The two cases can be physically identified with particle-particle (or hole-hole) and particle-hole correlations, respectively. Within this picture, it is also straightforward to see that $G_{\sigma\sigma'}(k, k')$ vanishes for non-interacting systems, as the two single-particle distributions are then statistically independent. Note that the converse of this statement does not hold.

In Fig. 4, we show the density-density correlation function $G_{\uparrow\downarrow}(k, k')$ in the first Brillouin zone at constant total particle number $N = 26$, for a variety of polarizations at the intermediate coupling $\gamma = -2.0$. For all results discussed in the following, we have fixed the spatial lattice size to $N_x = 64$ which we found sufficient to study relevant features (see, e.g., Ref. [23]).

The top-left panel of Fig. 4 shows the unpolarized case, for which the dominant correlations are expected to occur at the momentum space coordinates $(\pm k_F^{\uparrow}, \mp k_F^{\downarrow})$, reflecting BCS-type pairing, i.e., pairing at equal and opposite momenta between spin-up and -down particles. Since the Fermi momenta of the two species coincide in this case, the correlation peaks occur on the anti-diagonal $k = -k'$ and thus indicate pairing with a total pair-momentum of $q = k + k' = 0$.

For non-zero polarizations, a similar – yet distinct – picture emerges: While pairing still appears to happen close the respective Fermi points, the peaks of positive correlation shift "outwards" from the central anti-diagonal line by an offset of $|q|$. This is a hallmark of FFLO-type

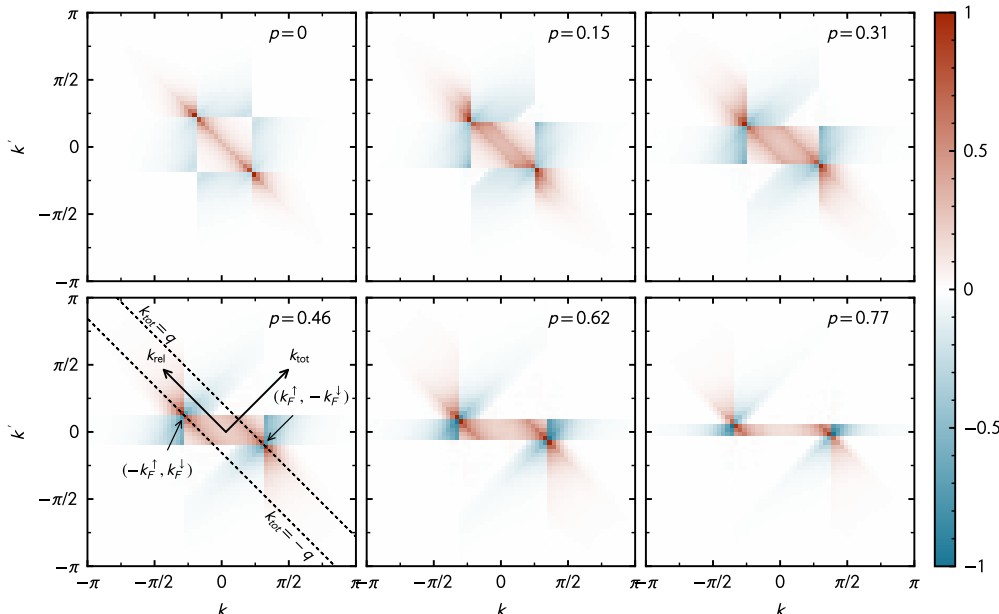

Figure 4: Shot-noise correlation function $G_{\uparrow\downarrow}(k, k')$ for polarized systems with $N_\uparrow + N_\downarrow = 26$ particles for configurations $(N_\uparrow, N_\downarrow) = (13, 13), (15, 11), (17, 9), (19, 7), (21, 5)$, and $(21, 3)$ corresponding to polarizations $p = 0, 0.15, 0.31, 0.46, 0.62$, and $0.77$ (from top left to bottom right). Positive (negative) correlations are indicated by red (blue) color coding, white color indicates that there is no correlation. To facilitate a better comparison between the different particle numbers, all values have been normalized to the respective maximal absolute values of the shot-noise such that the range of the correlation is always in the interval $[-1, 1]$. **Lower left panel**: The direction along the momentum-space diagonal defines the total momentum $k_{\text{tot}}$ of a pair whereas the direction along the anti-diagonal defines the relative momentum $k_{\text{rel}}$. The two dashed lines show lines of constant $k_{\text{tot}} = \pm q$ which include the points of opposite Fermi momenta labeled with their respective momentum-space coordinates.

correlations which we already discussed in the previous section. Since the positive particle-particle correlations dominate over the negative particle-hole contributions, singlet-pairing clearly dominates other ordering mechanisms, in particular the formation of a charge-density wave.

Another interesting feature is the "checkerboard" pattern, that separates the Brillouin zone into segments of positive and negative correlations at the intersection lines of the Fermi points. Employing the argument above, we expect four particle-hole-like areas with negative correlations alongside with five particle-particle (or hole-hole) regions with positive correlation. This is reflected in our numerical data which is also in excellent agreement with the observations in Ref. [23].

With increasing interaction strength, it becomes energetically more and more favorable to scatter particles from well below the Fermi point which leads to a broadening of the peaks of positive correlation at the opposite Fermi points. In unpolarized systems, this broadening eventually leads to the vanishing of the checkerboard pattern whereas the pattern is extremely stable in the spin-imbalanced case, indicating the "preservation" of the respective Fermi points as expected for systems with gapless excitations [75].

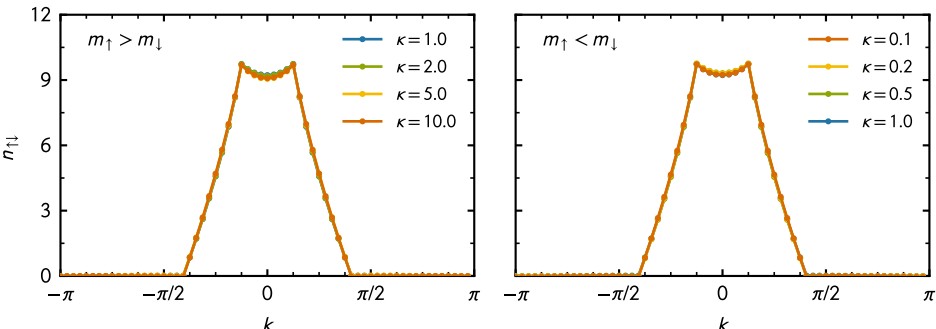

Figure 5:    Pair-momentum distribution $n_{\uparrow\downarrow}$ for mass-imbalanced systems with $(N_\uparrow, N_\downarrow) = (17,9)$ particles on a lattice of $N_x = 64$ sites and $\tilde{\gamma} = -2.0$. **Left panel**: Heavy-majority systems with mass ratios $\kappa = 1.0, 2.0, 5.0, 10.0$. **Right panel**: Heavy-minority systems with mass ratios $\kappa = 0.1, 0.2, 0.5, 1.0$. The statistical error bars are smaller than the symbol size.

# 4   Spin-polarized fermions with unequal masses

Turning now to the spin- and mass-imbalanced case, we note that the reduced symmetry of the Hamiltonian renders the system non-integrable, such that numerical methods have to be employed. Several studies based on DMRG [49], MC [47] and TEBD [48] have explored the phase diagram for either purely mass-imbalanced or spin-imbalanced 1D mixtures before. In this section, we present results for mass ratios which are in the range of potentially realizable values in experiments. This should allow for a check of previously made predictions on the stability of pairing in the presence of a finite mass imbalance in 1D Fermi mixtures.

In the following we set the scales such that the mass $m_l$ of the lighter species is fixed to $m_l = 1$. This leaves the mass $m_h$ of the heavier fermion species at our disposal, which we characterize by the numerical factor $\kappa$ as $m_h = \kappa m_l$ (see also Sec. 2.1 for our conventions). For the balanced case, we thus obtain a reduced mass $\mu = \frac{1}{2}$ and hence $\tilde{\gamma} = \gamma$, whereas the reduced mass increases for mass ratios $\kappa \neq 1$ from $\mu = \frac{1}{2}$ up to a maximum of $\mu = 1$ in the limit of $\kappa \to \infty$. Keeping $\tilde{\gamma}$ fixed ensures that we take out any trivial dependence on the mass imbalance. Any residual dependence of our results on the mass imbalance should be a true many-body effect.

As mentioned above, we now have to deal with a reduced symmetry of the Hamiltonian. To be specific, because of the additional mass asymmetry, the system is no longer invariant under a flip of the sign of the polarization, $p \to -p$. This requires us to consider two separate scenarios: The case where the majority species is heavier than the minority (heavy-majority case) and the opposite case (heavy-minority case). We keep the convention as in the spin-polarized case, namely $N_\uparrow > N_\downarrow$. This implies that $\kappa > 1$ ($\kappa < 1$) corresponds to a heavy-majority (heavy-minority) system.

## 4.1   Pair-momentum distribution

As a first step, we investigate again the diagonal of the pair-momentum distribution $n_{\uparrow\downarrow}(k,k)$ as we did in the previous section. In Fig. 5, we show $n_{\uparrow\downarrow}(k,k)$ for $(N_\uparrow, N_\downarrow) = (17,9)$ particles on a lattice of $N_x = 64$ sites, corresponding to a density of $n \approx 0.41$. Interestingly, we observe almost no variation of $n_{\uparrow\downarrow}(k,k)$ as a function of $\kappa$ for both the heavy-majority case (left panel) and the heavy-minority case (right panel). Most importantly, for all mass imbalances considered in this work, we observe constant positions of the maxima, indicating the stability of the FFLO-type correlations in the presence of a finite mass imbalance. Of course, the robustness

of the pair-momentum distribution with respect to a variation of the mass imbalance translates directly into a robustness of the on-site pair-correlation function, which is the Fourier transform of the pair-momentum distribution.

For the considered mass ratios, which are in the range of potentially realizable values, we observe FFLO-type correlations, regardless of the spin polarization. At the interaction strength considered here, $\tilde{\gamma} = -2.0$, this is in line with previous studies [47–49].

For mass imbalances beyond those considered in the present work, it has been observed in the asymmetric 1D Hubbard model that FFLO-type ordering is eventually suppressed. Instead, a charge-density-wave-type phase appears before either phase separation sets in [45] or the system collapses [41,47]. The "critical imbalances" above which these transitions occur depend on the interaction strengths but are expected to be well above the experimentally realizable values for the mass ratio $\kappa$. In any case, an exploration of this highly mass-imbalanced regime is beyond the scope of the present study [1].

For completeness, we comment on the possible occurrence of multi-particle bound states at large mass imbalances. For commensurate spin imbalances (i.e. integer ratios of the spin-up and spin-down densities), stable phases – where ordering is associated with the formation of such multi-particle bound states – have been observed [43, 44, 49, 81]. For general spin imbalances away from the commensurate points, however, such phases are unstable and are thus not relevant for the present discussion.

## 4.2 Shot-noise correlations

It may appear counterintuitive that a finite mass imbalance seems to leave the pairing of spin-up and spin-down particles unchanged altogether, as suggested by our results for the pair-momentum distribution. Loosely speaking, by increasing the mass of a fermion species, the spacing between the energy levels decreases (or, in terms of the Hubbard model, the bands flatten with decreased hopping while retaining the same number of states). With such a rescaled energy spectrum, the participation of higher-lying momentum states in scattering processes appears energetically possible and therefore pairing should also take place (far) away from the Fermi points. This is indeed the case. As we shall see below, however, pairs still predominantly form with the FFLO-momentum $q = |k_F^\uparrow - k_F^\downarrow|$ even in the mass-imbalanced case, such that only certain combinations of momenta of the spin-up and spin-down particles away from the Fermi points are preferred. In essence, the energy in mass-imbalanced systems is still minimized by accommodating the excess particles at the almost gapless points (i.e. at the nodes of the on-site pair-correlation function) which implies pairing at finite $k_{tot}$. In the following we analyze this aspect with the help of the shot-noise correlator $G_{\uparrow\downarrow}$ which shows distinct fingerprints of this mechanism.

Recently, studies based on ED explored shot-noise correlations [82] and a related measure [83] to study the influence of mass imbalance on pair formation in harmonically trapped spin-balanced few-body systems have also been performed. In the present work, however, we address this quantity in the presence of spin imbalance for many-body systems and show that unambiguous features emerge which can potentially be identified in currently available experimental setups.

---

[1] In [47], a detailed investigation for the onset the collapsed state in the Hubbard model was performed. For a (rough) translation of our parameters to the ones in [47], we may relate the Hubbard model at low filling to Eq. (1) via the kinetic energies $t \sim \frac{1}{2m}$ and therefore $U/t \sim 2mg$. For $\kappa = 10.0$ (which is the largest value considered in this work and on the upper end of the current capabilities of our CL implementation) and the interaction strength $\tilde{\gamma} = 2.0$ the ratio $t_2/|U| \sim 1/(2g\kappa) \approx 0.083$, placing our results well above typical values where the collapse occurs for any polarization studied (see Fig. 4 in [47]).

### 4.2.1 Heavy-majority systems

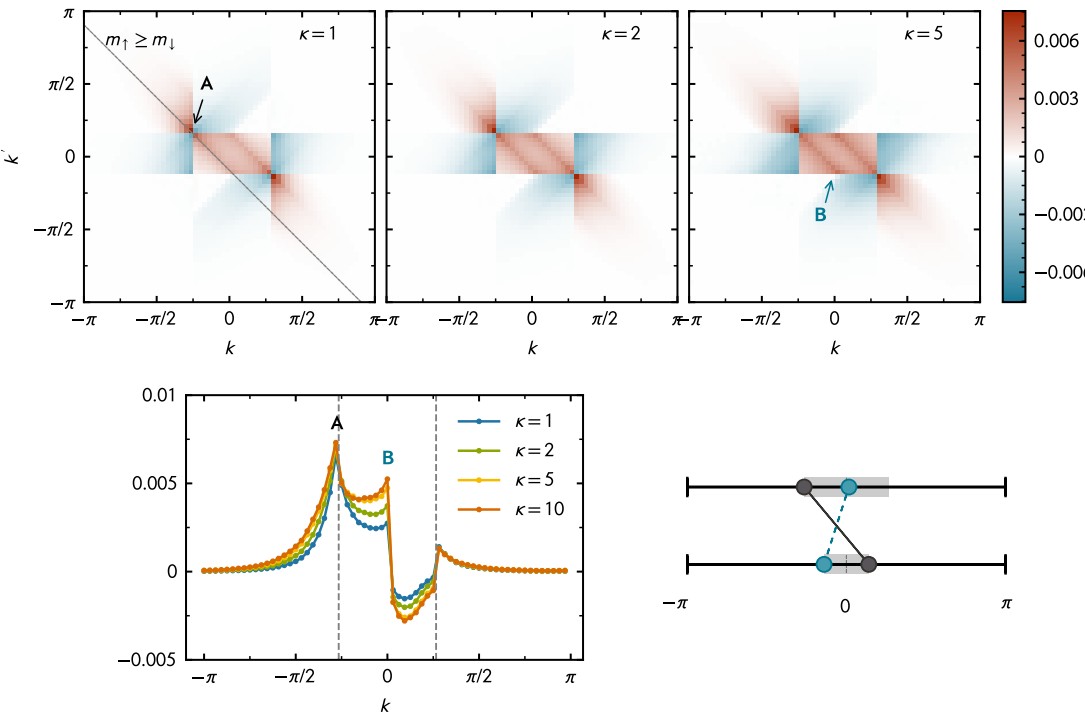

Figure 6: Pairing in heavy-majority systems. **Top row panels**: Shot-noise correlator $G_{\uparrow\downarrow}$ for $(N_\uparrow, N_\downarrow) = (17, 9)$ particles and mass imbalances $\kappa = 1, 2$, and 5. **Bottom left panel**: Cuts of $G_{\uparrow\downarrow}$ along constant $k_{\text{tot}} = -q$, corresponding to the gray dashed line in the upper left panel (the statistical error bars are smaller than the symbol size). The peaks labeled "A" and "B" correspond to the points marked in the upper panels. The vertical gray dashed lines indicate the position of the Fermi points associated with the majority species $k = \pm k_F^\uparrow$. **Bottom right panel**: Sketch of the pairing mechanism in momentum space. The upper (lower) line depicts the first Brillouin zone of the spin-up (spin-down) species. The end points of the gray-shaded bands correspond to the Fermi points of the two species. Finally, the big black dots connected with a black solid line symbolize standard BCS-type pairing "across the Fermi lines" (gray-shaded bands) which gives rise to peak A. The big blue dots connected with a dashed blue line symbolize the second most dominant pairing correlation which gives rise to the peak B.

First, let us consider the heavy-majority scenario in more detail, for which we show our results in Fig. 6. As discussed above, in the balanced case (top-left panel), pairing predominantly occurs at lines of constant momentum $k_{\text{tot}} = \pm q = \pm|k_F^\uparrow - k_F^\downarrow|$, which corresponds to $k' = k \pm q$ and describes the positions of the two maxima in the pair-momentum distribution discussed above. We observe peaks at $(\pm k_F^\uparrow, \mp k_F^\downarrow)$ which are associated with the expected FFLO-type pairing in the vicinity of both Fermi points. The dominant peak labeled with "A" in the top-left panel of Fig. 6 is located on the line $k_{\text{tot}} = -q$.[2]

With increasing mass ratio $\kappa$, additional peaks emerge at $(\pm k_F^\uparrow \mp 2q, \mp k_F^\downarrow)$, indicated by

---

[2]In addition, an equally dominant peak on the line $k_{\text{tot}} = q$ exists. For our discussion below, however, it is sufficient to focus on the line corresponding to $k_{\text{tot}} = -q$. Indeed, since the shot-noise correlator $G_{\uparrow\downarrow}(k, k')$ is invariant under a point reflection with respect to the origin $k = k' = 0$, it suffices to consider the shot-noise correlator on one side of the anti-diagonal defined by $k' = -k$.

stronger particle-particle correlations (i.e., darker red coloring). For the sake of readability, only the point at $(-k_\text{F}^\uparrow + 2q, k_\text{F}^\downarrow)$ is labeled with "B" in the rightmost panel of Fig. 6. The emergence of a peak at the coordinates $(k_\text{F}^\uparrow - 2q, -k_\text{F}^\downarrow)$ reflects pairing of light fermions "sitting" close to the opposite Fermi point with heavy fermions whose momenta are shifted such that the total momentum remains fixed.

The build-up of a second peak is highlighted in detail in the bottom-left panel of Fig. 6 where we show the cut along the line of $k_\text{tot} = -q$ (corresponding to the dashed line in the upper-left panel) for different mass imbalances. The configurations corresponding to the peaks are the ones that are most likely to be found. Of course, this does not mean that fermion pairs with a different internal momentum structure are not formed in the system. The probability to find the latter pairs is just lower.

The bottom-right panel of Fig. 6 illustrates the underlying pairing mechanism: For sufficiently large mass imbalance, it becomes energetically more favorable to pair up with heavy particles from far below the Fermi point while still obeying the constraint $|k_\text{tot}| = |q|$. The sketch depicts the perspective of the lighter (spin-down) particle at the Fermi point: In addition to forming a pair with a spin-up particle at $k = -k_\text{F}^\uparrow$ (big black dots), there is now a significant probability to pair up with a particle at $k_\uparrow = -k_\text{F}^\uparrow + 2q$ (big blue dots).

### 4.2.2 Heavy-minority systems

The investigation of the heavy-minority scenario, which is shown in Fig. 7, reveals a similar picture albeit with reversed roles of the majority and minority particles. For the same mass ratios as above, we now observe additional peaks at $(\pm k_\text{F}^\uparrow, \pm k_\text{F}^\downarrow \mp 2k_\text{F}^\uparrow)$, already clearly visible as additional "fringes" in the middle and the right panel of Fig. 7. The build-up of a secondary peak becomes again more clearly visible by comparing the shot-noise correlations along the line $k_\text{tot} = -q$, see bottom-left panel of Fig. 7. The additional peak "C" emerges at the expense of some weight at the opposite Fermi point of the majority species.

The underlying pairing mechanism is depicted in the bottom-right panel of Fig. 7 where pairing of a light majority particle sitting at its Fermi point is now favored to occur with either a heavy-minority particle sitting at its Fermi point (big black dot) or a fermion far above the Fermi point (big red dot). Similar to the heavy-majority case, the latter option becomes energetically more favorable with increasing mass ratio since higher-lying states can, loosely speaking, be reached more easily. Recall that the difference between the energy levels is proportional to the inverse of the mass of a particle. The corresponding peak in the shot-noise correlation function is labeled with "C" in the top-right panel of Fig. 7.

## 5 Discussion & Summary

To summarize, we have investigated the ground-state pairing behavior of non-relativistic spin- and mass-imbalanced Fermi gases in 1D, which is expected to be of the FFLO type. To that end, we employed the complex Langevin method (applied to the present system for the first time) and calculated two-body correlation functions, finding remarkable agreement with earlier theoretical predictions and numerical calculations, wherever available. In particular, we have observed hallmark features of the FFLO pairing across all spin polarizations considered in agreement with earlier studies on related systems.

Most importantly, we have presented results for the density-density correlation function. The latter shows an exceptionally clean signal for pairing across the mismatched Fermi points and is also accessible in experimental setups. Especially for spin- and mass-imbalanced systems, our study has revealed remarkable features with increasing mass imbalance. In fact, we

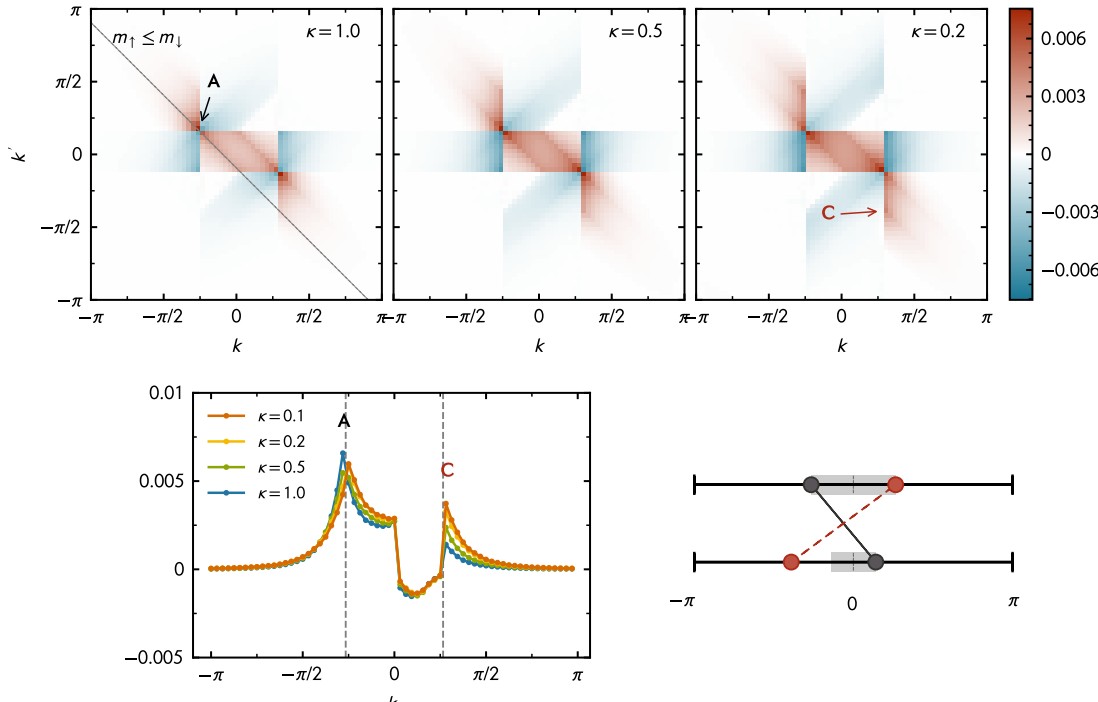

Figure 7: Pairing in heavy-minority systems. **Top row panels**: Shot-noise correlator $G_{\uparrow\downarrow}$ for $(N_\uparrow, N_\downarrow) = (17, 9)$ particles and mass imbalances $\kappa = 1, 0.5$ and $0.2$ (from left to right). **Bottom left panel**: Cuts of $G_{\uparrow\downarrow}$ along constant $k_{\text{tot}} = -q$, corresponding to the gray dashed line in the upper left panel (the statistical error bars are smaller than the symbol size). The peaks labeled "A" and "C" correspond to the points marked in the upper panels. The vertical gray dashed lines indicate the position of the Fermi points associated with the majority species $k = \pm k_F^\uparrow$. **Bottom right panel**: Sketch of the pairing mechanism in momentum space. The upper (lower) line depicts the first Brillouin zone of the spin-up (spin-down) species. The end points of the gray-shaded bands correspond to the Fermi points of the two species. Finally, the big black dots connected with a black solid line symbolize standard BCS-type pairing "across the Fermi lines" (gray-shaded bands) which gives rise to peak A. The big red dots connected with a dashed red line symbolize the second most dominant pairing correlation which gives rise to the peak C.

have found indications that, even for very large mass imbalances, the dominant correlations are of the FFLO type. However, in addition to the naively expected conventional FFLO-type pairing-peaks appearing at the opposite Fermi points in the density-density correlation function, we have observed that new peaks emerge with increasing mass imbalance. Interestingly, the spectral weight is found to be only shifted along anti-diagonals of constant total momentum of the fermion pairs. This is consistent with our finding of a constant peak position in the pair-momentum distribution. In particular, we have found that the aforementioned new peaks build up at the Fermi points of the light fermion species, pushing some of the weight of the heavier component away from its own Fermi point. Phenomenologically, the effect may be understood by recognizing that it is "easier" to move the higher-mass component to a different momentum state since the energy expense for this process decreases with increasing fermion mass.

It should be noted that technically the finding of a peak in the pair-correlation function (as

well as the corresponding features in the shot-noise) alone does not constitute a proof that the FFLO-type pairing is indeed the leading instability - nevertheless, it may be taken as a strong indication. In combination with what is known in the literature for the studied range of parameters, however, we are confident that the FFLO-type pairing is indeed the leading instability for the presented results. To unambiguously address this point, a precise determination of the correlation exponents is required. However, this is beyond the scope of the present study and is therefore deferred to a subsequent investigation.

While the present work focuses on bulk systems, i.e., we have employed periodic boundary conditions, we nevertheless expect it to be of relevance for experiments where the particles reside in non-uniform trapping potentials. In fact, earlier studies of the shot-noise for trapped systems suggest the stability of the distinct pairing patterns in the presence of harmonic (and likely any other) confinement [23], and even in the few-body sector [82, 84].

The success of the CL method in computing two-body correlation functions sets the stage for further investigations based on this method. In the future, it will be interesting to study the temperature dependence of the pairing patterns found here. In addition, future efforts will be directed towards systems in higher spatial dimensions, where long-range order is more stable against fluctuations. A particularly exciting avenue concerns the advent of quantum gas microscopes, which give direct access to shot-noise correlation functions for two-dimensional systems in optical lattices and could provide a viable way to study, e.g., the fate of pair correlations across the BCS-BEC crossover in 2D spin- and mass-polarized Fermi gases.

# Acknowledgements

The authors acknowledge numerous fruitful and enlightening discussions with H.W. Hammer, J.R. McKenney, and A.G. Volosniev. J.B. acknowledges support by the DFG under grant BR 4005/4-1 (Heisenberg program). J.B. and L.R. acknowledge support by HIC for FAIR within the LOEWE program of the State of Hesse. This material is based upon work supported by the National Science Foundation Computational Physics Program under Grant No. PHY1452635. Numerical calculations have been performed on the LOEWE-CSC Frankfurt.

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
