# Peer review of "Pairing patterns in one-dimensional spin- and mass-imbalanced Fermi gases"

_SciPost Physics, doi:SciPost Phys. 9, 014 (2020)_

## Round 1 · Referee Report · Anonymous (Referee 1) · 2020-4-20

Strengths

  1. The manuscript is based on accurate calculations based on the complex Langevin approach, which was initially developed for high energy physics and is now used to tackle condensed matter problems. The authors show that the method competes well with DMRG and quantum Monte Carlo for one dimensional systems and might be a reference in higher dimensions.

  2. The manuscript addressed the problem of superconducting pairing analysis in systems with combined spin and mass imbalance. While the 1D FFLO pairing have been largely discussed in literature, the authors found that the mass imbalance allows for a subdominant pairing mechanism, where the light fermion is at the Fermi surface and the heavy one is either deep inside it or deep outside it. This result is quite non trivial.

Weaknesses

I don't see significant weakness.

Report

The present manuscript investigate the ground state properties of spin-1/2 attractive fermions with combined spin and mass imbalance. The microscopic model is the generalization of the Gaudin model to unequal fermionic masses and is not integrable. The authors employed a novel numerical technique based on complex Langevin dynamics, which was initially developed in the context of high energy physics, to study noise correlations of the Fermi gas. These quantities can be measured experimentally with cold atoms systems.

The main claim of the manuscript is that, in the presence of combined spin and mass asymmetry, Cooper pairs with finite center-of-mass momentum q not only form between up and down fermions situated at the Fermi surfaces, but new forms of (subdominant) FFLO pairing arise. In particular a light fermion at the Fermi surface can pair with a down fermion which is either well inside the Fermi surface or well outside, depending on whether the up fermion component is majoritary or not.

I find the manuscript very interesting. There are some points that the authors should address to improve the quality of the presentation.

  1. In the discussion of Fig.1 there is an ambiguity between the physical density n of the gas entering the definition of gamma=g/n, and the lattice density, that is the filling factor corresponding to the spatial discretization. The latter should be small to mimic the original continuum system. In Fig.1 how the different data points for a fixed value of gamma are obtained ? By changing both the filling factor and the value of g so that the ratio g/n is kept constant ?

  2. The definition of the center of mass momentum of the pair k_cm=(k+k’)/2 seems inconsistent with the notation used afterwards. For a Cooper pair undergoing FFLO pairing (k+k’)= k_Fup-k_Fdown=q implying k_cm=q/2, while in Fig.4 and afterwards it is noted k_cm=q. The notation should be consistent.

  3. The dominant correlation in 1D attractive spin-imbalanced systems is FFLO superconducting pairing, which is a well established fact. On which basis also the subdominant correlations discussed in the manuscript are attributed to superconducting pairing rather than to a charge density wave ? Because the q vector is the same as for standard FFLO ? This point is unclear to me.

  • validity: top
  • significance: top
  • originality: high
  • clarity: high
  • formatting: excellent
  • grammar: excellent

Author:  Lukas Rammelmüller  on 2020-07-02  [id 869]

(in reply to Report 1 on 2020-04-20)
Category:
answer to question

We thank the referee for her/his positive comments. We have adapted our manuscript accordingly and hope that the referee recommends publication of the present version of our manuscript in SciPost.

1) We believe that the ambiguity between density and filling factor originates from the fact that we fix the lattice spacing to $l = 1 (a.u.)$ in our calculations (see Sec. 2.2), such that the density and filling factor assume the same numerical value. The results for different densities shown in Fig. 1 are obtained for varying particle numbers on appropriately sized grids, such that the filling factor (and hence the density) is fixed to $v = 0.5$. A variation of the coupling then yields the dependence on the dimensionless coupling $\gamma$.

2) We thank the referee for pointing out this inconsistency. We have fixed the appropriate pieces in the manuscript.

3) As it stands, we do not have a definite proof of our statement. It is rather a strong indication based on the analysis of the spatial modulation of the (quasi) order parameter, i.e., the hallmark property of the FFLO-type state. For example, we can obtain the spatial modulation from a Fourier transformation of the pair-momentum distribution: If pairing occurs at the same center-of-mass momentum with and without mass imbalance, then the spatial modulation of the order parameter will also be dominated by the same frequencies. In other words, the peak of the pair-momentum distribution does not change its position, which is presented in Fig. 5 of the manuscript. We believe that this argument is backed up by the fact, that our choice of parameters is well below the threshold to the collapsed state, i.e., where earlier studies (of the Hubbard model, see Ref. [47]) have found dominating superfluid correlations (for further details on the latter matter, we kindly refer to the answer to the second referee report, as well as the newly added footnote on page 12). To clarify this point, we have added an appropriate comment in Section 5 (at the end of the third-to-last paragraph; "It should be noted that technically ... to a subsequent investigation."). Moreover, we have slightly adapted our wording on page 15 (line 7 from below) where we now explicitly use the word "indications" to make clear that we do not have a proof at this point.

---

## Round 1 · Referee Report · Anonymous (Referee 2) · 2020-5-5

Strengths

1) Very readable presentation of materials with good introductory materials (both methods and physics) 2) Method used: Complex Langevin approach is relatively novel: this has been benchmarked here against Bethe Ansatz results where available (equal mass case for ground state energies). 3) Observables calculated here are indeed experimentally relevant as of now, and show what can be experimental signatures for eg. FFLO state. 4) Unexpected sub-leading correlations found in the density-density correlation function in momentum space.

Weaknesses

Nothing.

Report

The complex Langevin method has been used to numerically compute pair-momentum distribution and density-density correlations in momentum space, to characterise the FFLO state in a 1D continuum system of 2 species of fermions ("spin-1/2") with imbalances in both numbers of each species, and mass of each species. (The spin-imbalance only case has been much studied.) The main results showing the signatures and stability of the FFLO state against mass-imbalance (for not too large imbalances) are the continuum analogues of previous results on the Hubbard model (eg Ref. 47 in paper). One main novelty is the finding of (sub-leading) enhanced positive correlations between a fermion deep inside Fermi Sea and another near Fermi Surface (for heavy-majority system), or between fermions far outside of Fermi Sea and near Fermi Surface (for light-majority system), when mass-imbalance is increased strongly. This new finding came about from the new results on density-density correlations in momentum space.

These "atomic shot noise" results are certainly interesting, but before we can conclude definitively that the FFLO state is stable against not too strong mass-imbalances, can we check the following?

1) One main competitor for the FFLO state is some form of phase separation/collapse, and/or CDW (of pairs) state. While the stated scope of this paper does not not extend to huge mass-imbalances where collapse/CDW are seen, is it possible to translate roughly the mass-imbalances ratios used here to the hopping matrix elements imbalances ratios in the lattice case in eg. Ref 47? This way, we can see how far we may be from the collapse (CDW) etc. regimes. 2) In real optical lattice cold atom experiments, there is also a trapping potential that could favour some form of phase separation: eg centre region of (roughly) equal number of the two species and the outside regions with only the excess species. Ref. 49 has shown that this can change the relative stability of different phases for fermionic Hubbard model with spin and mass-imbalances. Can this happen also in the continuum case here? 3) What would a CDW state, or a collapsed state with some real space patches with CDW correlations (eg. in heavy-minority case cf Ref 47), look like in these atomic shot noise measurements? 4) Ideally, a direct computation of normal density-density correlation in real space (ie particle-hole analogue of particle-particle version here) should show that CDW is subheading to FFLO here?

Requested changes

1) Please add a color coding scheme for Fig. 4,5,7, to allow to see how strong the leading vs sub-leading correlations are. Currently, only some specific directions in (k',k) have been plotted (lower plots in Fig.6,7) that give indications for strength. 2) Fig. 3 dashed lines (not in the inset): what are they? Please state in Fig. caption or in main text?

  • validity: top
  • significance: high
  • originality: high
  • clarity: high
  • formatting: excellent
  • grammar: perfect

Author:  Lukas Rammelmüller  on 2020-07-02  [id 870]

(in reply to Report 2 on 2020-05-05)
Category:
answer to question

We thank the referee for the comments and questions regarding our manuscript to which we reply in detail below. We hope that the referee deems the present version of our manuscript suitable for publication in SciPost.

1) We want to thank the Referee for this suggestion. The new version of the manuscript includes a footnote (on page 12) with a (rough) translation of the physical parameters in order to facilitate an easier comparison between the present work and what has been presented in the literature before. To be precise, we have included a translation to the conventions used in Fig. 4 of Ref. [47], from which it is now visible that our results are well above the critical value of t2/|U| where the collapse is typically observed.

2) As fas as we know, phase separation in the presence of a harmonic trap is not specific to lattice setups and also occurs in continuum systems. In experiment, this is frequently observed not only in 1D systems (see, e.g., https://doi.org/10.1038/nature09393) but also in 3D systems (see, e.g., https://doi.org/10.1103/PhysRevLett.97.030401). The numerical exploration of this issue, while admittedly of great interest, is beyond the scope of the current implementation of our many-body method which in its present form requires periodic boundary conditions. Therefore, it is best suited for the study of spatially uniform systems (see also our reply to comments/questions 3) and 4) below).

3) + 4) Correlations for CDW ordering manifest themselves at $k^{\prime} = k \pm 2 k_F$ in the equal-spin shot-noise correlation and hence also in the total shot noise. For the mass-imbalanced Hubbard model, such correlations have been investigated in Refs. [23] and [75] of our list of references. In these studies it was found that the CDW correlations, are dominated by the FFLO correlations and are therefore subleading in this model. Since our model is similar to these systems (at least at low filling), we did not compute this quantity in the present work. However, we would expect an equivalent signature of this type of correlation also in the mass-imbalanced case. With respect to the collapsed state, we find it difficult to make a robust statement since we did not investigate this regime at all (see also the above). We agree that the question is very interesting but, at the same time, it is very challenging to tackle with our present setup: The implementation of the numerical method used in the current manuscript hinges on the spatial uniformity of the system. This hinders a study of this interesting regime. Still, it is worthwhile to note that an implementation including such inhomogeneous effects is in principle possible within the CL approach but has not been explored yet. In summary, we naturally find the study proposed by the referee interesting. However, an in-depth discussion of these effects would go well beyond the scope of the present work which aims at a first exploration of correlations functions in a spin- and mass-imbalanced environment with a numerical method that has not yet been employed in this context so far. More detailed studies are deferred to future work. To clarify this point and make our standpoint clear for the reader, we have also added a comment in Section 5 (at the end of the third-to-last paragraph; "It should be noted that technically ... to a subsequent investigation.").

Regarding the requested changes: 1) A suitable color-coding was added to the figures mentioned by the referee. 2) These are the non-interacting values, an appropriate comment has been added to the caption of Fig. 3.

---

## Round 2 · List of Changes

• fixed some inconsitencies w.r.t. the definition of the center-of-mass momentum / total momentum
  • a colorbar was added to Fig. 4, Fig. 6 and Fig. 7
  • a paragraph was added in Sect. 5, to further clarify the nature of our findings
  • a footnote was added in Sect. 4, to provide a (rough) translation to known parameter values in the literature

---

## Editorial Decision

published